# A Scalable Temporal-Spatial Framework for Transaction Anomaly Detection in Ethereum Networks

## Abstract

The rapid evolution of the Ethereum network necessitates sophisticated techniques to ensure its robustness against potential threats and to maintain transparency. While Graph Neural Networks (GNNs) have pioneered anomaly detection in such platforms, capturing the intricacies of both spatial and temporal transactional patterns has remained a challenge. This study presents a fusion of Graph Convolutional Networks (GCNs) with Temporal Random Walks (TRW) enhanced by probabilistic sampling to bridge this gap. Our approach, unlike traditional GCNs, leverages the strengths of TRW to discern complex temporal sequences in Ethereum transactions, thereby providing a more nuanced transaction anomaly detection mechanism. Extensive evaluations demonstrate that our TRW-GCN framework substantially advances the performance metrics over conventional GCNs in detecting irregularities such as suspiciously timed transactions, patterns indicative of token pump and dump schemes, or anomalous behavior in smart contract executions over time. As baseline algorithms for comparison, common unsupervised methods such as Isolation Forest, One-Class SVM, and DBSCAN (as classifier for TRW-GCN embedding) are employed; finally our novel TRW-GCN plus scoring method is compared with the state-of-the-art temporal graph attention algorithm.

## 1 Introduction

Graph Convolutional Networks (GCNs) have emerged as a transformative tool in the domain of graph-structured data representation. Their ability to encapsulate both local and global graph structures has paved the way for their application in diverse fields. However, as the scale and intricacy of graph data have surged, the efficient training of GCNs has become a paramount concern. Traditional training paradigms, although effective, are often encumbered by high computational and storage demands, especially when dealing with expansive graphs. The realm of GCN training has witnessed a burgeoning interest in sampling methods, particularly those rooted in probabilistic frameworks within graphs. Layer-wise sampling methods have been at the forefront of prior advancements. Chen et al. (2018) in their work on FastGCN championed the cause of probabilistic sampling on independent nodes. Their approach was further nuanced by Huang et al. (2018) which introduced the concept of layer-dependent sampling, thereby adding another dimension to the sampling process.

While traditional GCNs have shown remarkable potential in handling static graph structures, their application to dynamic graphs introduces new challenges and opportunities. In order to extend GCNs to dynamic graphs, it is crucial to understand how learning on dynamic graphs works, which is a relatively recent area of research. There have been studies which investigate discrete-time graphs represented as a sequence of graph snapshots (Yu et al., 2019; Sankar et al., 2020; Pareja et al., 2019; Yu et al., 2018), also several continuous-time approaches have been presented (Xu et al., 2020; Trivedi et al., 2019; Kumar et al., 2019; Ma et al., 2018; Nguyen et al., 2018; Bastas et al., 2019; Rossi et al., 2020), where continous dynamic graphs means that edges can appear at any time (Rossi et al., 2020). Liu et al. (2023) mentioned that most temporal graph learning methods model current interactions by combining historical information over time, however, such methods merely consider the first-order temporal information. To solve this issue, they proposed extracting both temporal and structural information to learn more informative node representations. In our ablation study, we focus mainly on TGAT by Xu et al. (2020) which has proved superior in performance.

Also, the topic of anomaly detection in Blockchain has received considerable attention. For example, in Ethereum, the unexpected appearance of particular subgraphs has implied newly emerging malware (Xu and Livshits 2019). Anomaly detection in blockchain transaction networks is an emerging area of research in the cryptocurrency community (Lee et al. 2022). Wu et al. (2020) investigated phishing detection in blockchain network using unsupervised learning algorithms. Ofori-Boateng et al. (2021) have also discussed topological anomaly detection in multilayer blockchain networks. Given that the Ethereum network witnesses dynamically evolving transaction patterns, it becomes imperative to account for the temporal sequences and correlations of transactions. While some literature has touched upon temporal networks, there is a conspicuous absence of comprehensive research that deeply integrates TRW with GCNs, and probabilistic sampling, especially within the blockchain environment. Furthermore, the specific challenge of anomaly and transaction burst detection in the Ethereum network, which has massive implications for network security and fraud detection, has not been extensively explored using these combined methodologies. As Ethereum continues to grow and evolve, addressing such gap with an appropriate methodology becomes increasingly crucial to ensure the security, scalability, and robustness of the network. This study addresses the pressing challenge of detecting time-sensitive anomalies within Ethereum blockchain transactions. We propose a novel approach, designed to provide both the spatial and temporal dynamics inherent in Ethereum transaction data. Our research offers several contributions:

**Enhanced Anomaly Detection with TRW** : Our model leverages TRW in tandem with GCN to improve anomaly detection effectiveness. By integrating temporal patterns, our approach can identify irregularities such as suspiciously timed transactions, patterns indicative of token 'pump and dump' schemes, or anomalous behavior in smart contract executions over time.

**Efficiency in Sampling Representative Nodes**: Given the substantial size and continuous growth of the Ethereum blockchain, efficient sampling methods are essential. Our GCN, trained with TRW nodes, provides a solution that balances accuracy with computational efficiency.

**Detecting Patterns Leading to Sophisticated Attacks**: Decentralized networks are vulnerable to sophisticated attacks, particularly those that exploit timing vulnerabilities such as front-running attacks. Our proposed GCN with TRW integration aims to detect complex patterns such as MEV bots, rapid buying or selling of assets, or others which are activities that can exhibit time-sensitive anomalies; manual inspection is then necessary for further investigation of the attack.

## 2 MODEL DESIGN

GCNs are a pivotal neural network architecture crafted specifically for graph-structured data. Through the use of graph convolutional layers, we seamlessly aggregate information from neighboring nodes and edges to refine node embeddings. In enhancing this mechanism, we incorporate probabilistic sampling, which proves particularly adept in analyzing the vast Ethereum network. The incorporation of Temporal Random Walks (TRW) adds a rich layer to this framework. TRW captures the temporal sequences in Ethereum transactions and not only focuses on nodes' spatial prominence but also considers the transactional chronology. Here, 'time' is conceptualized based on the sequence and timestamps of Ethereum transactions, leading to a dynamically evolving, time-sensitive representation of the network.

Here, graph is represented as $G = (V, E)$, where $V$ is the set of nodes (vertices) and E is the set of edges connecting the nodes. Each node $v_i$ in the graph is associated with a feature vector $F_i$, and $F \in \mathbb{R}^{|V| \times 4}$ represents a feature matrix of size 4. Aggregation is a process to combine the feature vectors of neighboring nodes using an adjacency matrix $A$ to capture graph connectivity. To enable information propagation across multiple layers, the graph convolution operation is performed iteratively through multiple graph convolutional layers (GCLs). The output of one layer serves as the input to the next layer, allowing the propagation of information through the network. The node representations are updated layer by layer, allowing information from neighbors and their neighbors to be incorporated into the node features. The parameters $W^l$ are learned during the training process to optimize the model's performance on a specific graph-based task. GCNs often consist of multiple layers, where each layer iteratively updates the node representations:

$$h_i^{(l)} = Activation \left( W^{(l)} \text{Aggregate} \left( h_j^{(l-1)} | j \in N(i) \right) \right) \tag{1}$$

Here, $h_i^{(l)}$ is the representation of node $i$ at layer $l$, and $h_j^{(l-1)}$ is the representation of neighboring node $j$ at the previous layer (*l-1*). The final layer is usually followed by a global pooling operation to obtain the graph-level representation. The pooled representation is then used to make predictions.

## 2.1 Incorporating TRW into GCN

The TRW-enhanced GCN creates a multidimensional representation that captures both the structural intricacies and time-evolving patterns of transactions. Such an approach requires meticulous mathematical modeling to substantiate its efficacy, and exploring the depths of this amalgamation can reveal further insights into the temporal rhythms of the Ethereum network.

**Temporal Random Walk (TRW)**

Given a node *i*, the probability *Pij* of moving to a neighboring node *j* can be represented as:

$$P_{\mathrm{ij}} = \frac{\omega_{\mathrm{ij}}}{\sum_k \omega_{\mathrm{ik}}} \tag{2}$$

where $\omega_{\mathrm{ij}}$ is the weight of the edge between node $i$ and $j$, and the denominator is the sum of weights of all edges from node $i$. In a TRW, transition probabilities take into account temporal factors. Let's define the temporal transition matrix $T$ where each entry $T_{ij}$ indicates the transition probability from node $i$ to node $j$ based on temporal factors.

$$T_{\mathrm{ij}} = \alpha \times A_{\mathrm{ij}} + (1 - \alpha) \times f(t_{\mathrm{ij}}) \tag{3}$$

Where: $A_{ij}$ is the original adjacency matrix's entry for nodes $i$ and $j$. $\alpha$ is a weighting parameter. $f_{ij}$ is a function of the temporal difference between node $i$ and node $j$. Given this temporal transition matrix $T$, a normalized form $\widetilde{T}$ can be used for a GCN layer:

$$\widetilde{T} = \widetilde{D}_T^{-1} T \tag{4}$$

Where $\widetilde{D}_T$ is the diagonal degree matrix of $T$. To incorporate the TRW's temporal information into the GCN, we can modify the original GCN operation using $\widetilde{T}$:

$$h^{(l+1)} = \sigma\left( \widetilde{D}_T^{-\frac{1}{2}} \widetilde{T} \widetilde{D}_T^{-\frac{1}{2}} h^{(l)} W^{(l)} \right) \tag{5}$$

## 2.2 Effect on Anomaly Detection

The embeddings from a GCN (post TRW influence) should be more sensitive to recent behaviors and patterns. When these embeddings are passed to a classifier, clustering and scoring algorithms like DBSCAN, OCSVM, ISOLATION FOREST, and LOF, anomalies that are based on recent or time-sensitive behaviors are more likely to stand out. In our work, the term "anomaly" refers to patterns that are statistically uncommon or divergent from the norm based on the features learned by our model. These uncommon patterns, while not definitively erroneous, are of interest because they deviate from typical behavior. In the context of Ethereum transactions, such deviations could potentially indicate suspicious activities, novel transaction patterns, or transaction bursts.

While we here provide insight and a mathematical proof, the true value of TRW in improving GCN for anomaly detection is empirical. We would need to compare the performance of GCN with and without TRW on a temporal dataset to see tangible benefits (see section 3.5 and appendix C). Here is how temporal weights are applied:

1. Node Features are weighted by time: When updating the node features through the matrix multiplication, nodes that are temporally closer influence each other more, allowing recent patterns to be highlighted.

2. Temporal Relationships are captured: The modified node features inherently capture temporal relationships because they aggregate features from temporally relevant neighbors.

3. Higher Sensitivity to recent anomalies: With temporal weighting, anomalies that have occurred recently will be more pronounced in the node feature space.

**Theorem 1:** Enhancement in effectiveness of anomaly detection using GCN through TRW Integration.

**Proof.**
At a fundamental level, anomaly detection is the task of distinguishing outliers from normal data points in a given feature space. If we have an anomaly score function $s : \mathbb{R}^d \to \mathbb{R}$, we can detect anomalies by: $s(v) > \theta$ Where $\theta$ is a threshold, and $v$ is a vector in the feature space.

A GCN produces node embeddings (or features) by aggregating information from a node's neighbors in the graph. Let's express this aggregation for a single node using a simple form of a GCN layer:

$$h_i^{(l+1)} = \sigma \left( \sum_{j \epsilon Neighbors(i)} W h_j^{(l)} \right) \qquad (6)$$

Where $h_i^{(l)}$ is the feature of node $i$ at layer $l$, and $W$ is the weight matrix.

**Incorporating TRW:** With a temporal random walk, the aggregation process is influenced by time, so the aggregation becomes:

$$h_i^{(l+1)} = \sigma \left( \sum_{j \epsilon Neighbors(i)} T_{ij} W h_j^{(l)} \right) \qquad (7)$$

Where $T_{ij}$ is the temporal transition probability from node $j$ to node $i$. Let's assume a node with an anomaly will have a different feature vector from the nodes without anomalies. For simplicity, let's use the Euclidean distance as the anomaly score: $s(v) = \|v - \mu\|$ where $\mu$ is the mean vector of all node features. Given a temporal anomaly (an anomaly that's influenced by recent events), using TRW will result in a modified feature vector for the anomalous node. Let's consider two scenarios:

1. GCN without TRW: For an anomalous node $n$, its feature vector is: $h_n = \sigma \left( \sum_j W h_j \right)$

2. GCN with TRW: For the same anomalous node $n$, it becomes: $h'_n = \sigma \left( \sum_j T_{nj} W h_j \right)$

If the anomaly is temporally influenced, then $h'_n$ should be significantly different from $h_n$ due to the weights introduced by $T_{nj}$ (see appendix A for weight cancellation). In the context of our anomaly score function: $s(h'_n) - s(h_n) > \delta$ where $\delta$ is a value indicating the sensitivity of the temporal context; we will use this later in our scoring method. If the anomaly is truly temporally influenced, this difference will be significant, and thus, the GCN with TRW will have a higher likelihood of detecting the anomaly. From the linear algebra perspective, the effect of TRW on a GCN for anomaly detection is evident in how node features are aggregated. The temporal weights (from $T_{ij}$) make the GCN more sensitive to temporal influences, making it more adept at detecting anomalies.

**Theorem 2:** TRW sampling maintains higher temporal consistency than traditional random walk sampling.
The TRW framework introduces a model where the transition probabilities between nodes in a graph are temporally adjusted. The probability of transitioning from node i at time t to node j at time t+1, denoted as $P_{ij(t,t+1)}$, is influenced by temporal proximity. This stands in contrast to traditional random walks, where transition probabilities are solely based on the static adjacency matrix of the graph. We define $T_{\text{TRW}}(t)$ as the transition matrix for TRW at time t, with each entry $T_{ij}(t)$ representing the probability of transitioning from node i to node j at time t. Conversely, $T_{\text{RW}}$ represents the transition matrix for a traditional random walk, with constant transition probabilities over time. We measure temporal consistency by examining the variation in transition probabilities over time, with TRW expected to exhibit lower variation due to its emphasis on temporal proximity. We continue the proof in appendix B.

**Theorem 3:** Improvement of GCN performance with probabilistic sampling in the context of random walk sampling.
see appendix C for the complete analytical proof.

## 3 EMPIRICAL ANALYSIS

GCNs have achieved state-of-the-art performance in various image recognition problems due to their ability to automatically learn hierarchical features from raw data. Here, we combine it with TRW to make embedding in the Ethereum network. We run the models on a MacBook Pro equipped with an Intel Core i9 processor, featuring 8 cores, speed of up to 4.8 GHz, and 30 GB of RAM.

### 3.1 DATASETS AND EXTRACTING NODE FEATURES

Creating a complete transaction graph for all Ethereum blocks would be a computationally intensive task, as it would involve processing and storing a large amount of data. However, in the supplemental material we provide the code to generate a transaction history graph for a range of blocks. We further need to incorporate spatial and temporal Node Features to capture temporal aspects more explicitly:

**incoming_value_variance**: Variance of the transaction values received by the node. This metric quantifies the spread or dispersion of incoming transaction amounts, providing insight into the consistency or variability of funds received. **outgoing_value_variance**: Variance of the transaction values sent by the node. **activity_rate**: The activity rate of a node represents the total number of transactions (both incoming and outgoing) divided by the duration (in terms of blocks). It indicates the frequency of interactions involving the node over a specific period. **change_in_activity**: The change in activity refers to the difference in the number of transactions of the current block compared to the previous block for a given node. This metric captures fluctuations or deviations in transaction behavior over consecutive blocks. **time_since_last**: Time since the last transaction involving the node, measured as the difference between the current block number and the block number of the node's most recent transaction. It provides insights into the recency of activity associated with the node. **tx_volume**: Total transaction volume associated with the node, calculated as the sum of incoming and outgoing transaction values. This metric represents the overall magnitude of financial transactions involving the node. **frequent_large_transfers**: Indicator variable identifying addresses engaged in frequent and large transfers. Nodes meeting specific thresholds for both transaction frequency and volume are flagged. **gas_price**: Additional feature relevant for MEV detection, representing the gas price paid for transactions. Gas price fluctuations can signal potential MEV activities such as frontrunning or transaction ordering strategies. **token_swaps**: Another feature for MEV detection, indicating involvement in token swaps or trades on decentralized exchanges (DEXs). Analysis of token swap transactions can reveal arbitrage opportunities or manipulative behavior by MEV bots. **smart_contract_interactions**: Feature identifying transactions interacting with known DeFi protocols or smart contracts. MEV bots may exploit vulnerabilities or manipulate protocol behaviors.

### 3.2 TRW-GCN COMBINED METHOD TO DETECT ANOMALIES

To apply graph convolutional layers to the blockchain data for aggregating information from neighboring nodes and edges, we'll use the PyTorch Geometric library. This library is specifically designed for graph-based data and includes various graph neural network layers, including graph convolutional layers. Note that training and testing a graph neural network on Ethereum dataset would require significant computational resources, as currently, the Ethereum network possesses about 20 million blocks, which are connected over the Ethereum network, and we provide the transaction history graph within a specified block range.

In Algorithm 1, we intend to compare the anomaly detection of full- and sub-graphs (sampling using TRW). The graph convolution operation combines the features of neighboring nodes to update the representation of a given node. As node features, we input the 10 features indicated in 3.1 as vector representation; considering 20 hidden layers, 100 epochs, `lr=0.01`, `num_walks=10`, and `walk_length=100`, the resulting output vector aggregates information from all neighboring nodes. By using the nodes from TRW for training, the GCN will be more attuned to the time-dependent behaviors, leading to better detection of sudden spikes in transaction volume or unusual contract interactions that occur in quick succession. In our experiments, we employ TRW to sample nodes from the entire graph, ensuring that the graph's integrity is maintained. Here's how it can be done:

| Algorithm 1: TRW- GCN combined method to detect anomalies |
| --- |
| **Steps:** |
| 1. Load and Preprocess the graph $G$. |
| 2. Node feature extraction for each node $v_i \in V$: Construct a node feature matrix $F \in \mathbb{R}^{|V| \times 4}$ where each row $F_i$ corresponds to $f(v_i)$. |
| 3. Convert graph to adjacency matrix $A \in \mathbb{R}^{|V| \times |V|}$. |
| 4. Instantiate two GCN models $M_{TRW}$ and $M$ with parameters in_channels, hidden_channels, out_channels. |
| 5. Temporal Random Walk (TRW) for $k = 1$ to num_walks: Aggregate all walks in a set $W = \bigcup_{k=1}^{\text{num\_walks}} w_k$. |
| 6. Training using sampled-graphs: Train $M_{TRW}$ or $M$ using node features $F_N$ and adjacency matrix $A_N$. |
| 7. Anomaly Detection: Apply DBSCAN, One-Class SVM, IsoForest, and LOF on embeddings from the trained GCN model $M$ to obtain anomaly labels. |

| Algorithm 2: A Score-based anomaly detection associated with time-dependent behaviors |
| --- |
| **Steps:** |
| 1. Graph Preprocessing: $G' = G(V, E)$ where $E$ has node attributes. |
| 2. Node Feature Extraction: $X = [x_1, x_2, \ldots, x_n]$ for $n \in V$. Adjacency Matrix $A$ from $G'$. |
| 3. GCN Model: GCNModel with layers: in_channels $\rightarrow$ hidden_channels $\rightarrow$ out_channels. |
| 4. Temporal Random Walk: $\text{TRW}(G', \text{start}, \text{length})$ returns walk $W$ and timestamps $T$. |
| 5. Node Sampling via TRW: All_Walks $= \bigcup_{i=1}^{\text{num\_walks}} \text{TRW}(G', \text{random\_node}, \text{walk\_length})$. |
| 6. Node Frequency Computation: $\text{freq}(v) = \frac{\text{occurrences of } v \text{ in All\_Walks}}{\text{max occurrences in All\_Walk}}$ for $v \in V$. |
| 7. Anomaly Score Computation: $S(v) = \frac{(\text{emb}(v)_{\text{latest}} - \mu(\text{emb}(v)))}{\sigma(\text{emb}(v))} \times \text{freq}(v)$ where emb is the node embedding, $\mu$ is the mean, and $\sigma$ is the standard deviation; where anomalous nodes $v$ are where $S(v) > \text{threshold}$. |

1. **Perform TRWs to Sample Nodes for Training:** The TRWs provide sequences of nodes representing paths through the Ethereum network graph. Nodes appearing frequently in these walks are often involved in recent temporal interactions.

2. **Train the GCN with the Sampled Nodes:** Instead of using the entire Ethereum network graph for training, use nodes sampled from the TRWs. This approach tailors the GCN to recognize patterns from the most temporally active parts of the Ethereum network.

Using the GCN with TRW combined method, one can achieve 1) anomalies Detected, 2) Training Efficiency, and 3) Quality of Embedding. The integration of TRW with GCNs offers a novel approach for generating embedding that capture both spatial and temporal patterns within the Ethereum network. These embedding are vital for understanding the underlying transaction dynamics and for effectively detecting anomalous activities. To evaluate the potential of the TRW-GCN methodology, we employ four distinct machine learning techniques: DBSCAN, SVM, Isolation Forest (IsoForest), and Local Outlier Factor (LOF). Wu et al. (2020) indicated that they have obtained more than 500 million Ethereum addresses and 3.8 billion transaction records. However, only 1259 addresses are labeled as phishing addresses collected from EtherScamDB, which implies an extreme data imbalance as the biggest obstacle for phishing detection, therefore they used unsupervised learning detection method. We similarly use unsupervised learning for detection in our GCN-TRW algorithm.

The extensive use of these four diverse techniques allows us to validate the efficacy of the TRW-GCN framework. The high anomaly detection rates in Figure 1 by clustering methods underscores the importance of algorithm selection. As observed in Figure 1, these techniques seem to be sensitive to the embedding generated by TRW-GCN, as the number of anomalies vary significantly with and without TRW. It's essential to note that high detection doesn't necessarily imply high precision; it might indicate a higher false positive rate in the ML methods, clustering methods like dbscan in particular, as also shown in Table 1. Nevertheless, Figure 1 vividly showcases the superiority of the TRW-GCN combined approach over traditional GCN with higher anomaly detection. The enhanced detection capabilities can be attributed to the TRW's ability to encapsulate temporal sequences and correlations of transactions. It is more interesting to find out which node feature mainly contributes to anomaly detection, we show it in Figure 2. As illustrated by different colors, the feature 3-6 namely activity_rate, change_in_activity, time_since_last (mainly the temporal features) are the drivers of frequent anomalies (with dark blue colors), while tx_volume and frequent_large_transfers (with green colors) also produce anomalies but less frequently. Although we have obtained good insights into the method effectiveness to detect time-dependent patterns and features, but we should look for more precise and less prone to error detection method.

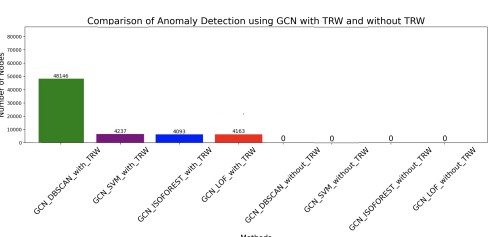

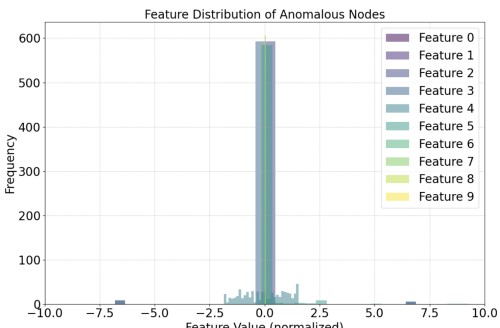

Figure 1: A comparison of 4 detection models namely dbscan, svm, isoforest and lof between full-graph and sub-graph with TRW sampling in 1000 blocks which include 83252 nodes, and 101403 edges.

Figure 2: Feature distribution where Blue and Green colors: activity_rate, change_in_activity, and times_since_last show highest frequencies.

Table 1: Comparison of Precision/Recall/F-score of 4 methods with/out-TRW

| Method | Prec.(w-T) | Recall(w-T) | F-score(w-T) | Prec.(o-T) | Recall(o-T) | F-score(o-T) |
|---|---|---|---|---|---|---|
| DBSCAN | 0.497 | 0.758 | 0.600 | 0.450 | 0.75 | 0.600 |
| SVM | 0.499 | 0.636 | 0.599 | 0.450 | 0.601 | 0.546 |
| IsoForest | 0.515 | 0.051 | 0.093 | 0.512 | 0.051 | 0.093 |
| LOF | 0.487 | 0.049 | 0.088 | 0.470 | 0.047 | 0.087 |

### 3.3 SCORE-BASED ANOMALY PATTERN

As seen in Table 1, the 4 methods provide relatively low precision (we do not obtain the precision by one-class SVM reported in Wu et al. 2020) and While traditional methods compute anomaly scores based on the relative position or density of data points in the feature space, we need a method to be more focused on temporal dynamics, tracking the evolution of each node's embedding over time and weighing it by the node's frequency in the graph. To adapt the code to pick up anomalous patterns associated with time-dependent behaviors, the algorithm should be equipped to recognize such patterns. First augment the node features to capture recent activity, with time features as explained in dataset section. After obtaining node embedding from the GCN, compute the anomaly score for each node based on its temporal behavior. The simplest way to achieve this is by computing the standard deviation of the node's feature over time and checking if the latest data point deviates significantly from its mean.

In the integrated code, Algorithm 2, we altered the node features to capture recent activity. After training the GCN and obtaining the embedding, we then compute an anomaly score based on how much the recent transaction volume (the latest day in our case) deviates from the mean. We then use a visualization function to display nodes with an anomaly score beyond a certain threshold (in this case, we've used a z-score threshold of 2.0 which represents roughly 95% confidence).

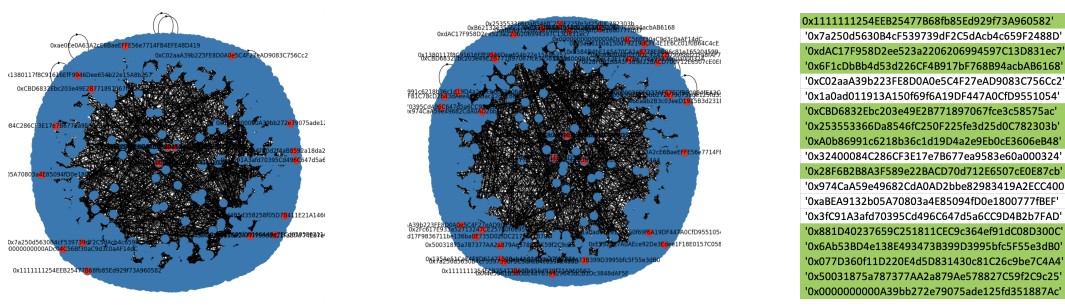

Figure 3: Anomaly detection in (left) 100 blocks with 6 features, (middle) 100 blocks with 10 features, (right) the anomalous addresses where the time-sensitive associated ones are hashed green.

In Figure 3, black points represent the vast majority of nodes in the Ethereum network dataset. They signify regular non-anomalous Ethereum addresses. Cluster of points inside and around the blue circle represent groupings of Ethereum addresses or contracts that have had frequent interactions with each other. The density or proximity of points to each other indicates how closely those addresses or contracts are related. Red points would represent the nodes that have been flagged as anomalous based on their recent behavior. The code identifies them by computing an anomaly score, and those exceeding a threshold are colored red. In the left graph, there are just 20 nodes detected as anomaly in 100 blocks where we used 6 structural features in our detection algorithm, while in the right graph, we used 10 features to detect anomalies in the same 100 blocks, and 12 more suspicious addresses are detected, hashed in green. This signifies the importance of temporal feature selection, as by adding 4 temporal features we would be able to detect missing anomalies. We checked these addresses in Ether explorer website https://etherscan.io , and found the corresponding labels such as MEV Bot, Metamask: Swap, Uniswap, Wrapped Ether, Rollbit, Blur: Bidding, which are mainly time-sensitive transactions or contracts, see next section for further explanation on what is normal versus anomaly. In Table 2, we explain the types of detected anomalies.

Table 2: Some types of detected anomalies

| Detected anomalous patterns | Explanation |
| --- | --- |
| MEV Bot (Miner Extractable Value) like ————— 0x6F1cDbBb4d53d226CF4B917bF768B94acbAB6168 | MEV strategies can affect the fairness and efficiency of the Ethereum network, and certain MEV activities may be considered harmful. |
| Uniswap (users to swap various ERC-20 tokens) like 0x3fC91A3afd70395Cd496C647d5a6CC9D4B2b7FAD, and Metamask Swap router like ————— 0x881D40237659C251811CEC9c364ef91dC08D300C | Uniswap smart contracts facilitate decentralized token swaps and are not inherently anomalous; Large-scale token swaps on Uniswap could be used for trading strategies or liquidity provision. |
| Flashloan (borrowing a large sum of tokens and repaying the loan within the same block. or Blur: Bidding 0x0000000000A39bb272e79075ade125fd351887Ac | Detecting flash loans typically involves monitoring for transactions with large token volumes and analyzing their timing and patterns. |

### 3.4 NORMAL VERSUS ANOMALY, BASELINE ALGORITHM, ALGORITHM COMPLEXITY, AND THE GROUND TRUTH

In Ethereum, what may be considered normal or anomalous behavior can vary depending on various factors such as market conditions, network activity, and the specific use cases of different addresses or smart contracts. Time-sensitive irregularities in Ethereum transactions refer to anomalies that occur within specific time frames or exhibit patterns that are indicative of immediate or rapid actions. These irregularities may include instances of rapid buying or selling of assets, front-running other traders, MEV activities, flash loan exploits, or token swaps executed within short time intervals. Identifying these irregularities requires analyzing transactional data in real-time or within narrow time windows to capture anomalous behaviors as they occur. See Table 3 for a list of time-sensitive items in Ethereum network including transactions, contracts, and platform activities. Our objective is to identify such instances. These transactions represent potential threats to the integrity and fairness of the Ethereum network, necessitating further investigation and scrutiny. Upon identifying suspicious transactions, our approach advocates for thorough investigation and validation. This involves cross-referencing transaction details with external sources such as Etherscan.io in Table 2, and employing manual review processes to assess the legitimacy of the flagged activities.

Similar to the papers by Wu et al. (2020), Feng et al. (2023), and Zhang et al. (2023), as baseline algorithms for comparison, common unsupervised methods such as Isolation Forest, One-Class SVM, and DBSCAN are employed. Evaluation metrics, including precision, recall, F1 score in Table 1 are utilized to assess the performance of the proposed method using training and test data. However, clustering methods seem to report many false positives, and we do not also obtain the precision reported by Wu et al. (2020). The study further introduces a statistically-based scoring method to identify anomalous nodes. The scoring function employs different z-score thresholds of 1.0, 1.5, and 2.0 (95% confidence level), and on average it produces the precision of 80%. Furthermore, we compare the results obtained from our scoring method with the ground truth on etherscan.io, providing a case-by-case evaluation of some detected time-sensitive anomalies in Table 2; all detected anomalies are re-affirmed with manual inspection.

Table 3: some time sensitive items on Ethereum network and their definitions

| Time sensitive items | Definitions |
|---|---|
| MEV Bot | MEV refers to the additional profit that miners can extract from the Ethereum network by reordering, censoring, or including transactions in blocks. The timing of transactions and block mining can affect the potential profit extracted by MEV bots. |
| Metamask: Swap Uniswap | Uniswap is a decentralized exchange (DEX) protocol on Ethereum, and swaps conducted through MetaMask can be time-sensitive, especially considering the volatility of cryptocurrency prices and liquidity on Uniswap. |
| Flashloan | Flash loans are uncollateralized loans that must be borrowed and repaid within a single transaction block. These loans are often used for arbitrage, liquidations, or other trading strategies that require rapid execution. |
| Wrapped Ether (WETH) | Wrapped Ether (WETH) is an Ethereum token pegged to the value of Ether (ETH). Transactions involving WETH can be time-sensitive, especially if they're related to trading, liquidity provision, or token swaps. |
| Rollbit | Rollbit is a cryptocurrency trading platform, and transactions conducted on the platform can be time-sensitive, particularly in the context of trading activities, order executions, and market conditions. |
| Token Launches and Airdrops | Token launches and airdrops often have predefined distribution schedules or timeframes during which users can claim or receive tokens. Missing these deadlines may result in loss of opportunities or benefits. |
| Smart Contract Exploits | Exploiting vulnerabilities in smart contracts often requires precise timing to execute malicious transactions before vulnerabilities are patched or mitigated. |

We further compare the TRW-GCN model against the state-of-the-art TGAT method, and during our experiments with TGAT, we encountered significant computational and performance challenges. TGAT is designed to leverage temporal information and attention mechanisms to capture the dynamic nature of graphs. Despite this sophisticated approach, we observed that TGAT resulted in higher computational costs, primarily due to its multi-head attention mechanism, which involves multiple passes of matrix multiplications and attention score computations. Furthermore, TGAT's reliance on temporal edge attributes added another layer of complexity, further increasing the computational burden. Despite TGAT's advanced capabilities, the accuracy of the model in the Ethereum network was found to be as high as 65%, significantly lower than the average of 80% we already achieved with the TRW-GCN embedding plus the scoring classifier. One possible reason for this discrepancy could be the sensitivity of TGAT to the quality and scale of temporal data which is quite a challenge in the Ethereum networks (see the supplemental material to process Ethereum data for TGAT). Furthermore, using blockchain addresses as users in TGAT make its essential indexing very difficult, whereas TRW seems to be working far better in such networks.

To accurately determine the complexity of TRW-GCN, it is essential to integrate the complexities of both the TRW and the GCN components. The complexity of the TRW involves initiating walks from nodes and stepping through neighbors up to a specified length, which can be quantified as $O(W \times LE \times D \log D)$, where $W$ represents the number of walks, $LE$ the length of each walk, and $D$ the average degree of nodes. The GCN complexity involves aggregating features from neighboring nodes and applying transformations through learnable parameters across $L$ layers. The aggregation complexity is proportional to the number of edges $E$, and the transformation involves matrix multiplications depending on the number of features $F$ and hidden units $H$, resulting in $O(L \times (E + N \times F \times H))$. When combining these two aspects, the overall complexity of TRW-GCN is expressed as $O(W \times LE \times D \log D + L \times (E + N \times F \times H))$. The dominant complexity terms typically relate to $E$ and $N$, particularly in dense graphs or when handling high-dimensional feature transformations. See further the proof in appendix D. As seen in the TRW-GCN complexity graph, Figure 4, GCN complexity scaling with graph size is far more than the TRW complexity.

### 3.5 HOW TRW IMPACTS ON GCN PERFORMANCE AS COMPARED TO TRADITIONAL RW

Let's delve into empirical justification on why TRW sampling could enhance the performance of GCNs, especially in temporal networks like Ethereum. For a detailed mathematical proof on the probabilistic sampling in GCN, you are invited to read appendix C. One issue with traditional random walks is the potential for creating "jumps" between temporally distant nodes, breaking the

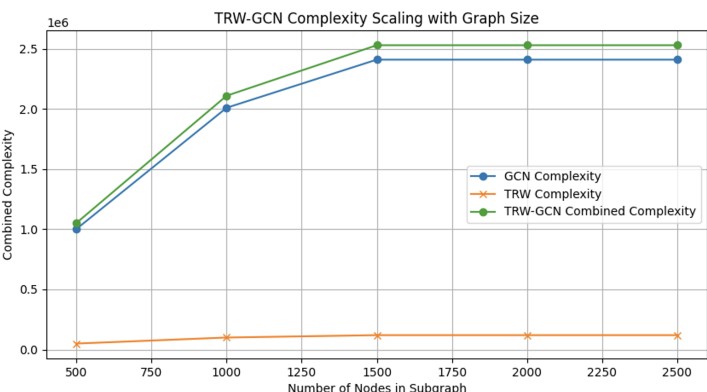

Figure 4: TRW-GCN complexity graph, where GCN complexity scaling with graph size is far more than TRW.

temporal consistency. GCNs rely on the local aggregation of information, and since TRW promotes smoother temporal signals, GCNs can potentially learn better node representations. Temporal consistency ensures that the sequences are logically and temporally ordered. This can be crucial for predicting future events or understanding time-evolving patterns, making GCNs more reliable. We compare different GCN models (including graphSAGE and graph attention network GAT model) for fullgraph, and sampled-graph with traditional and temporal random walk in Figure 5. Although one sees little difference between the accuracy of the fullgraph and the sampled-graph in graphSAGE and GAT models, one can see that traditional random walk and temporal random walk improve GCN accuracy, where TRW shows even further improvement than the traditional random walk.

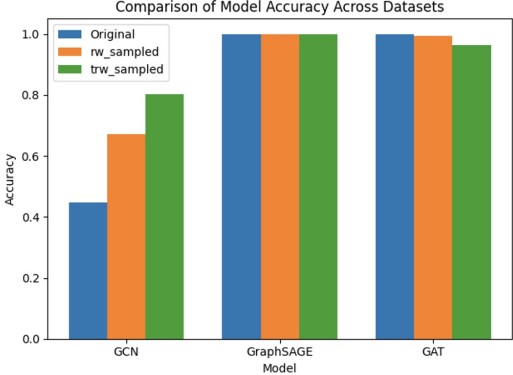

Figure 5: Comparison of accuracy of three GCN models between fullgraph, traditional RW and TRW-based on sampled graph in 100 blocks.

## 4 CONCLUSION

The evolution and complexity of the Ethereum network has heightened the urgency for temporal anomaly detection methods. Through our research, we've demonstrated that the convergence of GCNs and TRW offers a solution to this challenge. This fusion has enabled us to delve deeper into the intricate spatial-temporal patterns of Ethereum transactions, offering a refined lens for anomaly detection. We have shown the methodology usefulness by expressing and proving three distinct theorems, full empirical analysis and evaluation. While this approach is used to obtain the embedding, we have compared different clustering and scoring methods to obtain highest precision in anomaly detection, and verified with the ground truth found on etherscan.io. Furthermore, we have demonstrated the model that TRW-GCN improves anomaly detection, and proved how probabilistic sampling improves GCN performance.

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

# A  WEIGHT CANCELLATION IN THEOREM 1

In the topic of anomaly detection, particularly in systems where temporal factors play a crucial role, the design and behavior of the transformation matrix $T_{nj}$ are of paramount importance. This section delves into the potential challenges posed by weight cancellation within $T_{nj}$ and its consequential impact on the detection of temporally influenced anomalies (given in our first Theorem). We explore the nuances of how such weight cancellations can diminish the efficacy of anomaly detection and propose strategies to mitigate these issues. Additionally, we underscore the importance of empirical analysis in validating the robustness and reliability of our anomaly detection methodology under various scenarios. Let us first render our definitions:

**Vector Spaces**

Let $\mathbb{R}^m$ be the vector space of interest. Define $h_n \in \mathbb{R}^m$ as a feature vector in the absence of temporal influence. Let $T_{nj} \in \mathbb{R}^{m \times m}$ be a transformation matrix encoding temporal weights. Define $h'_n = T_{nj} h_n$, where $h'_n \in \mathbb{R}^m$ is the transformed feature vector under temporal influence.

Assume $T_{nj}$ has entries $t_{ij}$ where $i, j = 1, 2, \ldots, m$.

## DIFFERENCE MEASUREMENT

We use the Euclidean norm to quantify the difference: $||h'_n - h_n||_2$. This norm is given by

$$||h'_n - h_n||_2 = \sqrt{\sum_{i=1}^{m} (h'_{ni} - h_{ni})^2} \tag{8}$$

where $h'_{ni}$ and $h_{ni}$ are the components of $h'_n$ and $h_n$, respectively.

## EXPRESSION OF $h'_n$ IN TERMS OF $T_{nj}$ AND $h_n$

$h'_n = T_{nj} h_n$ implies

$$h'_{ni} = \sum_{j=1}^{m} t_{ij} h_{nj} \tag{9}$$

for each $i$.

## NORM CALCULATION

Compute the norm $||h'_n - h_n||_2$ as follows:

$$||h'_n - h_n||_2 = \sqrt{\sum_{i=1}^{m} \left( \sum_{j=1}^{m} t_{ij} h_{nj} - h_{ni} \right)^2}.$$

This equation represents the Euclidean norm of the difference between the transformed feature vector $h'_n$ and the original feature vector $h_n$.

## CONDITIONS FOR SIGNIFICANT DIFFERENCE

Given: $h'_n = T_{nj} h_n$ and $||h'_n - h_n||_2 > \epsilon$, for some threshold $\epsilon > 0$.

$$\text{For } ||h'_n - h_n||_2 > \epsilon, \text{ it must hold that } \sum_{i=1}^{m} \left( \sum_{j=1}^{m} t_{ij} h_{nj} - h_{ni} \right)^2 > \epsilon^2.$$

This inequality implies that, for at least one $i$, the inner sum $\sum_{j=1}^{m} t_{ij}h_{nj} - h_{ni}$ must be non-negligible. Therefore, the weights in $T_{nj}$ must be such that they do not merely scale $h_{ni}$ but rather significantly alter the distribution of $h_n$. Scaling would imply a uniform change across all components of $h_n$, which might not be sufficient to meet the inequality condition. Instead, the transformation must significantly alter the distribution of $h_n$. This could mean changing the relative magnitudes of its components, modifying their relationships, or introducing non-linear changes. Such alterations are necessary for effectively differentiating between normal and anomalous states, especially in the context of anomaly detection where temporal influences are considered.

SCENARIOS LEADING TO WEIGHT CANCELLATION

**Scenario Analysis:**

Consider the case where $T_{nj}$ has symmetric properties or specific patterns that lead to cancellation.

For instance, if $T_{nj}$ is such that $t_{ij} = t_{ji}$ for all $i, j$, and $h_n$ has symmetric properties, then

$$\sum_{j=1}^{m} t_{ij}h_{nj} \text{ could approach } h_{ni} \text{ for all } i. \tag{10}$$

Additionally, if $T_{nj}$ contains complementary weights, such as some $t_{ij}$ and $t_{ik}$ summing to zero, and $h_{nj}$ and $h_{nk}$ are similar, cancellation could occur.

**Analysis of $T_{nj}$ Properties for Cancellation**

To further understand how $T_{nj}$ might lead to weight cancellation:

- Consider the spectral properties of $T_{nj}$. If $T_{nj}$ has eigenvalues close to 1, then it acts close to an identity matrix on certain vectors.
- If $T_{nj}$ has orthogonal rows or columns, it might preserve the magnitude of $h_n$ under certain conditions, leading to minimal change in $h'_n$.
- If the entries of $T_{nj}$ are structured such that they negate each other when applied to $h_n$, this could lead to a scenario where $h'_n \approx h_n$.

In scenarios where the weights in the transformation matrix $T_{nj}$ cancel each other out, this can significantly impact the detection of temporally influenced anomalies. To mitigate these issues, several strategies can be employed:

**Regularization:** Introducing regularization is a good practice in preventing extreme weight values, which can be beneficial in any anomaly detection system, including Ethereum network analysis.

**Weight Initialization and Optimization:** Carefully initializing the weights in $\mathbf{T_{nj}}$ and employing robust optimization techniques can ensure that the weights evolve in a manner that minimizes the risk of cancellation. This can be particularly important in Ethereum network anomaly detection.

**Spectral Analysis:** Performing spectral analysis of $\mathbf{T_{nj}}$ to understand its eigenvalues and eigenvectors can provide insights into how the matrix behaves and identify potential scenarios where cancellation might occur. Adjustments can then be made accordingly.

**Ensemble Methods:** Using ensemble methods is a robust strategy in anomaly detection, as it reduces reliance on any single transformation. In the context of Ethereum network anomalies, ensemble methods can enhance the reliability of detection by combining multiple models or transformations.

## B   TRW SAMPLING MAINTAINS HIGHER TEMPORAL CONSISTENCY

**Theorem 2:** TRW sampling maintains higher temporal consistency than traditional random walk sampling.

**Definitions and Assumptions:**
- In TRW, the probability of transitioning from node $i$ at time $t$ to node $j$ at time $t + 1$ is given by $P_{ij}(t, t + 1)$, which is higher for temporally closer nodes.

- In a traditional random walk, the transition probability $P_{ij}$ is independent of time and is based solely on the adjacency matrix of the graph.

- Let $T_{\text{TRW}}(t)$ be the transition matrix for TRW at time $t$, where each entry $T_{ij}(t)$ represents the probability of transitioning from node $i$ to node $j$ at time $t$.

- Let $T_{\text{RW}}$ be the transition matrix for a traditional random walk, where each entry $T_{ij}$ is constant over time.

- Temporal consistency can be quantified by the variation in the transition probabilities over time. For TRW, this variation is expected to be lower than for traditional random walks, as TRW emphasizes temporal proximity.

**Proof:**

Consider the difference in transition probabilities between two consecutive time steps in TRW: $\|T_{\text{TRW}}(t + 1) - T_{\text{TRW}}(t)\|$. This norm is expected to be small, indicating high temporal consistency.

For a traditional random walk, the transition probabilities do not change over time: $\|T_{\text{RW}}(t + 1) - T_{\text{RW}}(t)\| = 0$. However, this does not imply temporal consistency, as it does not account for the temporal nature of the data.

Now we do comparison:

- To demonstrate higher temporal consistency in TRW, one can show that the variation in transition probabilities in TRW is more aligned with the temporal dynamics of the data compared to traditional random walk. This can be done by analyzing the correlation between $T_{\text{TRW}}(t)$ and the actual temporal sequence of events in the data.

- $T_{\text{TRW}}(t)$ aligns more closely with the temporal sequence of events than $T_{\text{RW}}$ and temporal consistency is better captured by a model that adjusts its transition probabilities based on the temporal proximity of events. Therefore, TRW is expected to maintain higher temporal consistency than traditional random walk sampling.

## C  IMPROVEMENT OF GCN PERFORMANCE WITH PROBABILISTIC SAMPLING

**Theorem 3:** Improvement of GCN performance with probabilistic sampling in the context of random walk sampling.

Providing a comprehensive mathematical proof on the theorem on improvement of GCN performance through probabilistic sampling in the context of analyzing the Ethereum network, even in a simplified scenario, is a complex task that requires careful consideration and detailed mathematical derivations.

Consider a simplified Ethereum transaction graph with N accounts (nodes), and M transactions (edges) between them. We aim to prove the performance improvement of a GCN using probabilistic sampling for the task of predicting account behaviors.

Assumptions:
1. Nodes (accounts) have features represented by vectors in a feature matrix X.
2. The adjacency matrix A represents transaction relationships between accounts.
3. Binary labels Y indicate specific account behaviors.

**Proof.**

### C.1  DEFINE THE GRAPH LAPLACIAN

Start with the definition of the normalized graph Laplacian $L = I - D^{-\frac{1}{2}} A D^{-\frac{1}{2}}$, where $D$ is the diagonal degree matrix and $A$ is the adjacency matrix.

## C.2 Traditional GCN performance

Derive the eigenvalues and eigenvectors of the Laplacian matrix L and show their significance in capturing graph structure. Derive the performance of a GCN trained on the full graph using these eigenvalues and eigenvectors:

Step 1: Deriving Eigenvalues and Eigenvectors of the Laplacian matrix $L$

Given the normalized graph Laplacian matrix $L$, let $\lambda$ be an eigenvalue of $L$ and $v$ be the corresponding eigenvector. We have $L_v = \lambda_v$. Solving for $\lambda$ and $v$, we get:

$$D^{-\frac{1}{2}} A D^{-\frac{1}{2}} v = (1 - \lambda) v \tag{11}$$

$$A D^{-\frac{1}{2}} v = (1 - \lambda) D^{\frac{1}{2}} v \tag{12}$$

This equation implies that $D^{-\frac{1}{2}} A D^{-\frac{1}{2}}$ is a symmetric matrix that is diagonalized by the eigenvectors $v$ with corresponding eigenvalues $1 - \lambda$. The eigenvectors and eigenvalues of $L$ capture the graph's structural information. Larger eigenvalues correspond to well-connected clusters of nodes in the graph, while smaller eigenvalues correspond to isolated groups or individual nodes.

Step 2: Deriving GCN Performance Using Eigenvalues and Eigenvectors

Now let's consider a scenario where we're using a GCN to predict node labels (such as predicting high-value transactions) on the full graph. The GCN's propagation rule can be written as:

$$h^{(l+1)} = f(\hat{A} h^{(l)} W^{(l)}) \tag{13}$$

where $h^{(l)}$ is the node embedding matrix at layer $l$, $f$ is an activation function, and $\hat{A} = D^{-\frac{1}{2}} A D^{-\frac{1}{2}}$. is the symmetrically normalized adjacency matrix, and $W^{(l)}$ is the weight matrix at layer l. The key insight is that if we stack multiple GCN layers, the propagation rule becomes:

$$h^{(L)} = f(\hat{A} h^{(L-1)} W^{(L-1)}) = f(\hat{A} f(\hat{A} h^{(L-2)} W^{(L-2)}) W^{(L-1)}) \ldots \tag{14}$$

We can simplify this as:

$$h^{(L)} = f\left( \hat{A}^{(l)} h^{(0)} W^{(0)} \prod_{l=1}^{L-1} W^{(l)} \right) \tag{15}$$

Using the spectral graph theory, we know that $\hat{A}^{(l)}$ captures information about the graph's structure up to L-length paths. The eigenvalues and eigenvectors of $\hat{A}^{(l)}$ indicate the influence of different sampled-graphs of length L on the node embeddings. Larger eigenvalues correspond to more significant graph structures that can impact the quality of learned embeddings. By leveraging the spectral insights, GCNs can focus their learning on graph structures that matter the most for the given task. In the case of probabilistic sampling, the convergence of eigenvalues signifies that the sampled graph retains essential structural information from the full graph. This implies that by training GCNs on $\hat{A}_{\text{sampled}}$, we are effectively capturing the key graph structures necessary for accurate predictions. This incorporation of spectral properties aligns the GCN's learning process with the inherent characteristics of the graph, resulting in improved performance. The embeddings learned by the GCN on the sampled graph become more indicative of the full graph's properties as the sample size increases, enabling more accurate predictions or more efficient training convergence.

## C.3 Probabilistic Sampling Approach

In this step, we'll introduce a probabilistic sampling strategy to select a subset of nodes and their associated transactions. This strategy aims to prioritize nodes with certain characteristics or properties, such as high transaction activity or potential involvement in high-value transactions. Assign a probability $p_i$ to each node i based on certain characteristics. For example, nodes with higher transaction activity, larger balances, or more connections might be assigned higher probabilities. For each node i, perform a random sampling with probability $p_i$ to determine whether the node is included in the sampled subset. Consider a graph with $N$ nodes represented as $N = \{1, 2, \ldots, N\}$. Each node $i$ has associated characteristics described by a feature vector $\mathbf{X}_i = [X_{i,1}, X_{i,2}, \ldots, X_{i,k}]$, where $K$ is the number of characteristics. Define the probability $p_i$ for node $i$ as a function of its

feature vector $\mathbf{X}_i$: $p_i = f(\mathbf{X}_i)$. Here, $f(\cdot)$ is a function that captures how the characteristics of node $i$ are transformed into a probability. The specific form of $f(\cdot)$ depends on the characteristics and the desired probabilistic behavior. For example, $f(\mathbf{X}_i)$ could be defined as a linear combination of the elements in $\mathbf{X}_i$:

$$p_i = \sum_{j=1}^{K} \omega_j X_{i,j} \tag{16}$$

Where $\omega_j$ are weights associated with each characteristic. The weights $\omega_j$ can be used to emphasize or downplay the importance of specific characteristics in determining the probability. After obtaining $p_i$ values for all nodes, normalize them to ensure they sum up to 1:

$$p_{\text{normalized}} = \frac{p_i}{\sum_{j=1}^{N} p_j} \tag{17}$$

Use the normalized probabilities $p_{\text{normalized}}$ to perform probabilistic sampling. Nodes with higher normalized probabilities are more likely to be included in the sampled subset, capturing the characteristics of interest. The specific form of $f(\cdot)$ and the choice of weights $\omega_j$ depend on the nature of the characteristics and the goals of the analysis. This approach allows for targeted sampling of nodes that exhibit desired characteristics in a graph.

### C.4 GRAPH LAPLACIAN FOR SAMPLE GRAPH

Given the sampled adjacency matrix $\hat{A}_{\text{sampled}}$, we want to derive the graph Laplacian $\hat{L}_{\text{sampled}}$ for the sampled graph. The graph Laplacian $\hat{L}_{\text{sampled}}$ is given by:

$$\hat{L}_{\text{sampled}} = I - \hat{D}_{\text{sampled}}^{-\frac{1}{2}} \hat{A}_{\text{sampled}} \hat{D}_{\text{sampled}}^{-\frac{1}{2}} \tag{18}$$

Where $\hat{D}_{\text{sampled}}$ is the diagonal degree matrix of the sampled graph, where each entry $d_{ii}$ corresponds to the degree of node $i$ in the sampled graph, and $\hat{A}_{\text{sampled}}$ is the sampled adjacency matrix.

$$d_{ii} = \sum_{j=1}^{N_{\text{sampled}}} \hat{A}_{\text{sampled},ij} \tag{19}$$

The modified Laplacian captures the structural properties of the sampled graph and is essential for understanding its graph-based properties.

### C.5 EIGENVALUE ANALYSIS AND CONVERGENCE

We derived

$$\hat{L}_{\text{sampled}} = I - \hat{D}_{\text{sampled}}^{-\frac{1}{2}} \hat{A}_{\text{sampled}} \hat{D}_{\text{sampled}}^{-\frac{1}{2}} \tag{20}$$

as the normalized graph Laplacian for the sampled graph. Let $\hat{\lambda}_i$ be the $i$-th eigenvalue of $\hat{L}_{\text{sampled}}$ and $\hat{v}_i$ be the corresponding eigenvector. We have

$$\hat{L}_{\text{sampled}} \hat{v}_i = \hat{\lambda}_i \hat{v}_i \tag{21}$$

The goal is to compare the eigenvalues of $L$ with the eigenvalues of $\hat{L}_{\text{sampled}}$ and show convergence under certain conditions.

**Theoretical Argument:**

As the sample size $N_{\text{sampled}}$ approaches the total number of nodes $N$ in the original graph, $\hat{L}_{\text{sampled}}$ converges to $L$. This implies that the eigenvalues of $\hat{L}_{\text{sampled}}$ converge to the eigenvalues of $L$.

1. **Step-wise convergence:**

   For simplicity, we'll denote the entries of $\hat{L}_{\text{sampled}}$ as $\hat{l}_{\text{sampled}}(i, j)$ and the entries of $L$ as $l(i, j)$. To prove the convergence, we need to show that $\hat{l}_{\text{sampled}}(i, j) \to l(i, j)$ as $N_{\text{sampled}} \to N$ for all $i$ and $j$.

2. **Eigenvalue convergence:**

   Once establishing that each entry of $\hat{L}_{\text{sampled}}$ converges to the corresponding entry of $L$, one can use this result to prove the convergence of eigenvalues. Eigenvalues are solutions to the characteristic equation of the matrix, which depends on its entries. If all entries of $\hat{L}_{\text{sampled}}$ converge to those of $L$, the characteristic equations of both matrices will be similar.

**Stochastic Convergence:** The convergence argument relies on the concept of stochastic convergence. As the sample size becomes large, the sampled graph's properties approach those of the original graph. This includes the behavior of the eigenvalues.

**Graph Structure Alignment:** The convergence occurs when the sampled subset of nodes is representative enough of the entire graph. This means that the sampled graph captures the structural characteristics that contribute to the eigenvalues of L. Under the assumption of sufficient representativeness and with a large enough sample size, the eigenvalues of L^sampled converge to the eigenvalues of L.

## C.6 CONVERGENCE OF GCN EMBEDDINGS

Recall that the graph convolutional operation can be expressed as

$$h^{(l+1)} = f(\hat{A} h^{(l)} W^{(l)}) \tag{22}$$

The spectral properties of $\hat{A}$ and $L$ are determined by the eigenvalues. As shown in the previous steps, as the sample size increases, the eigenvalues of $\hat{L}_{\text{sampled}}$ converge to those of $L$. Graph convolutional layers rely on the eigenvectors and eigenvalues of $\hat{A}$. The graph convolution operation $\hat{A} h^{(l)} W^{(l)}$ involves these spectral properties.

**Convergence of GCN Layers:** Because the eigenvalues of $\hat{A}$ and $L$ are converging, the impact of multiple graph convolutional layers on $h$ and $h_{\text{sampled}}$ becomes increasingly similar as the sample size increases.

1. **Layer-by-layer impact:**

   As we stack multiple graph convolutional layers, each layer applies the graph convolution operation sequentially. This means that the impact of each layer depends on the eigenvalues of $\hat{A}$.

2. **Convergence influence:**

   As the eigenvalues of $\hat{A}$ converge to those of $L$ due to the increasing sample size, the behavior of the graph convolutional layers on $h$ and $h_{\text{sampled}}$ becomes increasingly similar. The convergence of eigenvalues indicates that the structural characteristics of the sampled graph are aligning with those of the original graph. The graph convolutional layers are sensitive to these structural properties, and as the structural properties become more aligned, the impact of these layers on embeddings $h$ and $h_{\text{sampled}}$ will also become more aligned.

Since the top eigenvectors correspond to the major variations in the graph, as the spectral properties converge, the embeddings $h$ and $h_{\text{sampled}}$ learned by the GCN should increasingly align in terms of the top eigenvectors. As the eigenvalues of the Laplacian matrices converge, the behavior of the graph convolution operation and the resulting embeddings in both the original and sampled graphs becomes more similar. This implies that the embeddings learned by a GCN on the sampled graph $h_{\text{sampled}}$ will converge to the embeddings learned on the full graph $h$.

## C.7 IMPACT OF EIGENVECTOR ALIGNMENT ON GCN PERFORMANCE

Recall that the eigenvalues and eigenvectors of the Laplacian matrix capture the graph's structural properties. Eigenvectors corresponding to larger eigenvalues capture important patterns and variations in the graph.

**GCN Performance Analysis:**

1. **Predictive Power of Eigenvectors:** The alignment of top eigenvectors suggests that the information captured by these eigenvectors is consistent between the original and sampled graphs.

2. **Prediction Task:** If the prediction task relies on features that align with the graph's structural patterns, then the embeddings learned on the sampled graph will capture similar patterns as those on the full graph.

3. **Performance Convergence:** As the embeddings $h_{\text{sampled}}$ approach $h$ in terms of the top eigenvectors, the predictive performance of the GCN on the sampled graph should approach the performance on the full graph. The alignment of top eigenvectors implies that the information encoded in the embeddings learned by a GCN on the sampled graph converges to that of the embeddings learned on the full graph. This suggests that as $h_{\text{sampled}}$ converges to $h$, the predictive performance of the GCN on the sampled graph should approach that on the full graph, assuming the prediction task is influenced by the graph's structural patterns captured by these eigenvectors. However, the precise impact will depend on the nature of the graph, the quality of the sampling strategy, and the specific prediction task. To prove the improvement of GCN performance with probabilistic sampling, consider the following steps:

   (a) **Original Graph Performance (Without Sampling):** Let $E$ be the performance measure (e.g., accuracy) of the GCN trained and evaluated on the full graph $G$ using embeddings $h$, denoted as $E_{\text{full}}$.

   (b) **Sampled Graph Performance (With Probabilistic Sampling):** Now, consider the performance of the GCN trained and evaluated on the sampled graph $G_{\text{sampled}}$ using embeddings $h_{\text{sampled}}$, denoted as $E_{\text{sampled}}$.

   (c) **Improved Performance:** $E_{\text{sampled}} > E_{\text{full}}$ indicates an improvement in performance due to probabilistic sampling.
      - Utilize the previously shown argument: As the sample size increases, the embeddings $h_{\text{sampled}}$ converge to $h$ in terms of top eigenvectors.
      - With the alignment of top eigenvectors and the graph convolutional layers' convergence, the learned embeddings become more similar.
      - The improved alignment of embeddings captures more relevant structural information, potentially leading to improved prediction accuracy or other performance metrics.

By leveraging the convergence of embeddings and the improved alignment of top eigenvectors through probabilistic sampling, we can argue that the performance of the GCN on the sampled graph $G_{\text{sampled}}$ is expected to be better (higher accuracy, faster convergence, etc.) than on the full graph $G$. This proof highlights the positive impact of probabilistic sampling on enhancing the performance of GCNs in analyzing complex graphs like the Ethereum network.

# D    COMPLEXITY ANALYSIS OF TRW-GCN

To properly calculate and explain the complexity of a Temporal Random Walk Graph Convolutional Network (TRW-GCN) system, it's crucial to factor in both the GCN and the TRW parts in a cohesive manner. This combination entails not only the graph convolutions but also the dynamic aspect introduced by temporal random walks. Here's a refined approach to describing and computing the combined complexity.

- **Dependence on Graph Size:** The complexity shows linear dependence on the number of edges $E$ and nodes $N$, with an additional logarithmic factor related to the maximum node degree $D$.
- **Scalability:** The method scales well with larger graphs, though high-degree nodes $d$ can introduce additional complexity due to the sorting step in TRW.

## GCN COMPLEXITY

The complexity of GCN operations is primarily influenced by the number of nodes $N$ and edges $E$ in the graph. The two main steps in a GCN layer are feature propagation and aggregation.

- **Feature Propagation:** Each node aggregates features from its neighbors, which involves accessing the adjacency matrix and node feature matrix. This step takes $O(E)$ time since each edge defines a neighbor relationship that needs to be processed.
- **Feature Transformation:** Multiplying the node feature matrix by a weight matrix. This step is $O(N \cdot F \cdot H)$ where $F$ is the number of input features and $H$ is the number of output features.

For a GCN with $L$ layers, the overall complexity is:

$$O(L \cdot (E + N \cdot F \cdot H))$$

TRW COMPLEXITY

The complexity of a temporal random walk depends on the length of the walk $W$ and the number of walks $R$. For each step in the walk, the algorithm looks at the neighbors of the current node.

- **Walk Initialization:** Starting a walk from a random node, which is $O(1)$.
- **Neighbor Selection:** Sorting the neighbors by timestamp and selecting the next node. The worst-case complexity for sorting neighbors is $O(D \log D)$, where D is the average degree of nodes.

For a walk of length $W$, the complexity is:

$$O(R \cdot W \cdot D \log D)$$

ANOMALY SCORE COMPUTATION COMPLEXITY

The computation of anomaly scores involves calculating the mean and standard deviation of node embeddings, followed by the z-score calculation.

- **Mean and Standard Deviation Calculation:** For $N$ nodes, each with an embedding of size $H$, this is $O(N \cdot H)$.
- **Z-Score Calculation:** For $N$ nodes, this is $O(N)$.

Overall, the complexity is:

$$O(N \cdot H)$$

Combining the complexities of all components:

- **GCN Complexity:** $O(L \cdot (E + N \cdot F \cdot H))$
- **TRW Complexity:** $O(R \cdot W \cdot D \log D)$
- **Anomaly Score Computation:** $O(N \cdot H)$

Assuming $F$, $H$, $L$, $R$, and $W$ are constants, the overall complexity simplifies to:

$$O(E + N \cdot D \log D)$$

