# OpenReview forum: "A Scalable Temporal-Spatial Framework for Transaction Anomaly Detection in Ethereum Networks"
_ICLR.cc/2025/Conference — ICLR 2025 Conference Withdrawn Submission_

### Official Review · Reviewer_u3dN · 2024-10-29

**Soundness:** 1
**Presentation:** 1
**Contribution:** 1
**Rating:** 3
**Confidence:** 4

**Summary:**

This paper investigates anomaly detection in Ethereum transaction network. To capture both spatial and temporal patterns in the graph, the authors propose to fuse GCN with temporal random walks. Specifically, the temporal transition matrix $T$ is calculated and utilized in GCN, which takes temporal factors into consideration. Then, three widely used anomaly detection models are employed to the learnt embeddings. The results prove the effectiveness of the proposed temporal random walk module.

**Strengths:**

1.	This paper investigates anomaly detection in Ethereum transaction network, which is an important topic.

2.	Different unsupervised fraud detection methods are verified on the learnt node representations.

3.	Different GNN backbones are tested in the experiments.

**Weaknesses:**

1.	The authors claim that their proposed model is a scalable temporal-spatial framework for anomaly detection in Ethereum transaction network. However, there is no sufficient experiments to support it. The used full graph only contains 80k nodes.

2.	Lots of key details are missing for the proposed model. For instance, the authors model the transition matrix $T$ with by weighing adjacent matrix $A$ and temporal difference between each node pairs. It is not clear how temporal difference is modeled.

3.	The scalability of the proposed model is bounded by the complexity of GNN architecture, since the authors only replace the adjacent matrix $A$ with transition matrix $T$. And, the authors did not propose any new techniques to improve it.

4.	There are lots of spatial temporal graph models that the authors did not compare and they also can be used in the scenario of fraud detection. Thus, the experiments are quite weak. Please refer to the references below.

5.	In Figure 4, it is not clear what the complexity means in the y-axis.

6.	The constructed features are somewhat heuristic. It is not clear whether these features play an importance role in the fraud detection task. What if we only use the structure information for fraud detection?

7.	More ablation studies should be given. Is it necessary to use the temporal information in the graph? What if we regard it as a static and apply existing graph fraud detection algorithms to the Ethereum transaction network?

References:

[1] Duan W, He X, Zhou Z, et al. Localised adaptive spatial-temporal graph neural network[C]//Proceedings of the 29th acm sigkdd conference on knowledge discovery and data mining. 2023: 448-458.

[2] Cheng D, Wang X, Zhang Y, et al. Graph neural network for fraud detection via spatial-temporal attention[J]. IEEE Transactions on Knowledge and Data Engineering, 2020, 34(8): 3800-3813.

**Questions:**

1.	The authors claim that their proposed model is a scalable temporal-spatial framework for anomaly detection in Ethereum transaction network. However, there is no sufficient experiments to support it. The used full graph only contains 80k nodes.

2.	Lots of key details are missing for the proposed model. For instance, the authors model the transition matrix $T$ with by weighing adjacent matrix $A$ and temporal difference between each node pairs. It is not clear how temporal difference is modeled.

3.	The scalability of the proposed model is bounded by the complexity of GNN architecture, since the authors only replace the adjacent matrix $A$ with transition matrix $T$. And, the authors did not propose any new techniques to improve it.

4.	There are lots of spatial temporal graph models that the authors did not compare and they also can be used in the scenario of fraud detection. Thus, the experiments are quite weak. Please refer to the references below.

5.	In Figure 4, it is not clear what the complexity means in the y-axis.

6.	The constructed features are somewhat heuristic. It is not clear whether these features play an importance role in the fraud detection task. What if we only use the structure information for fraud detection?

7.	More ablation studies should be given. Is it necessary to use the temporal information in the graph? What if we regard it as a static and apply existing graph fraud detection algorithms to the Ethereum transaction network?

References:

[1] Duan W, He X, Zhou Z, et al. Localised adaptive spatial-temporal graph neural network[C]//Proceedings of the 29th acm sigkdd conference on knowledge discovery and data mining. 2023: 448-458.

[2] Cheng D, Wang X, Zhang Y, et al. Graph neural network for fraud detection via spatial-temporal attention[J]. IEEE Transactions on Knowledge and Data Engineering, 2020, 34(8): 3800-3813.

**Details Of Ethics Concerns:**

NA.

---

### Official Review · Reviewer_1qBf · 2024-10-31

**Soundness:** 1
**Presentation:** 3
**Contribution:** 1
**Rating:** 3
**Confidence:** 4

**Summary:**

1. This model leverages TRW in tandem with GCN to improve anomaly detection effectiveness. By integrating temporal patterns, the approach can identify irregularities such as suspiciously timed transactions, patterns indicative of anomalous behavior in smart contract executions over time.
2. Given the substantial size and continuous growth of the Ethereum blockchain, efficient sampling methods are essential. The GCN, trained with TRW nodes, provides a solution that balances accuracy with computational efficiency.
3. Decentralized networks are vulnerable to sophisticated attacks, particularly those that exploit timing vulnerabilities such as front-running attacks. This proposed GCN with TRW integration aims to detect complex patterns such as MEV bots, rapid buying or selling of assets, or others which are activities that can exhibit time-sensitive nomalies; manual inspection is then necessary for further investigation of the attack.

**Strengths:**

1. The presentation is clear, and the methodology is easy to follow.

2. The theoretical proof is comprehensive.

**Weaknesses:**

1. All major anomaly detection baselines using GNNs are missing, such as TGAT, PC-GNN, Semi-GNN, Care-GNN, and others.

2. The structure of the context is confusing, as it mixes the methodology section with the experiments section.

3. Only one dataset is used for evaluation, raising concerns about the adaptability of this method to other datasets.

4. There are no cross-references available in the paper, making it difficult to navigate related content.

**Questions:**

1. What is the major difference between TRW-GCN and other temporal GNNs, such as TGAT, TGNN, or STGNN?

2. What is the model performance when using a pure random walk to replace TRW?

3. Why is TRW only suitable for the GCN backbone while failing with GAT and GraphSAGE backbones?

4. The AUC metric is important in the anomaly detection problem. What is the performance on this metric?

---

### Official Review · Reviewer_815E · 2024-11-01

**Soundness:** 2
**Presentation:** 2
**Contribution:** 2
**Rating:** 3
**Confidence:** 5

**Summary:**

This paper introduces a framework for detecting transaction anomalies within the Ethereum network, blending Graph Convolutional Networks (GCNs) with Temporal Random Walks (TRW) and probabilistic sampling. The authors compare the performance of the TRW-GCN model against traditional GCN and baseline unsupervised methods, with evaluations showing superior anomaly detection accuracy using the integrated TRW-GCN framework.

**Strengths:**

Strengths
Originality
The paper makes a attempt by combining TRW with GCN for Ethereum anomaly detection.

Quality
The use of TRW enables probabilistic sampling, which improves computational efficiency and ensures the model’s scalability for large-scale transaction graphs in the Ethereum network.

Clarity
This article explains its theoretical framework relatively clearly.

Significance
By addressing the temporal dynamics of transaction anomalies, this framework can advances blockchain anomaly detection.

**Weaknesses:**

1. The abstract does not specifically describe the innovations of the paper, which makes it difficult for readers to quickly grasp the core of the article from the abstract.
2. The paragraphs in the introduction seem to be pieced together, lacking coherence. I recommend the authors improve the readability of the paper.
3. Although the authors mention in lines 60-61 that the Ethereum network needs to consider temporal dynamics, I still do not understand the background regarding why temporal correlations must be considered in the Ethereum network.
4. Previous work has already shown many sampling-based GNNs, like literature [1], and there are already numerous works on anomaly/phishing scam detection in the Ethereum network based on random walks [2]. The TRW mentioned in section 2.1 does not seem to present any novelty. Unless the authors can clarify further, I think the novelty of this paper is not enough to be published in ICLR.
5. There are some writing errors: line 120 should refer to P_{ij}; the variables I and j in formulas (2) and (3) should both be in italics.
6. The physical meanings of V, E, and F in line 95 are not clearly described.
7. It is recommended that the authors include a link to a code repository in the paper and open-source the code, as this would not only enhance the transparency and reproducibility of the research but also allow other researchers and developers to further explore and improve upon this framework, leading to broader applications.

[1] Bai, Youhui, et al. "Efficient data loader for fast sampling-based GNN training on large graphs." IEEE Transactions on Parallel and Distributed Systems 32.10 (2021): 2541-2556.
[2] Yan, Chuyi, et al. "Aparecium: understanding and detecting scam behaviors on Ethereum via biased random walk." Cybersecurity 6.1 (2023): 46.

**Questions:**

Please refer to the Weaknesses section for specific points of concern and suggestions for improvement.

---

### Official Review · Reviewer_2uvU · 2024-11-04

**Soundness:** 3
**Presentation:** 2
**Contribution:** 1
**Rating:** 3
**Confidence:** 5

**Summary:**

The paper designs a GCN based framework to detect anomalies in Ethereum transaction networks by incorporating both the structure and temporal aspects of the transaction graph data. The main contribution of the work is introduction of temporal random walks (TRW) within the GCN framework to emphasize the time dependent behaviors. Empirical results show the effectiveness of the proposed method for anomaly detection in Ethereum networks.

**Strengths:**

1. This paper proposes to take temporal information into account in GCNs and as a matter of fact, temporal information is universal in the real-world.
2.  The ideas behind the framework are simple and make sense, the algorithm flowcharts are clear.

**Weaknesses:**

W1. I think the introduction of TRW in GCNs are not novel at all. The literature survey completely bypasses TimeSAGE [1] (citation below), which incorporates temporal random walks in GraphSAGE (a generalization framework for GCNs). The framework presented here seems to have extremely high resemblance to [1]. Therefore, I’d appreciate a better positioning of the current work against [1].
W2. The theorem statements are vague and require proper statements. For example, Th 1, “Enhancement ineffectiveness,” is quite vague and doesn’t really convey anything. Similarly, Theorem 3 feels incomplete. The words in the theorem statements are not analytically defined. I’d appreciate a better presentation of the theorems.
W3. The choice of hyperparameters are arbitrary e.g. walk length and num_walks etc. How sensitive are the results with respected to these hyperparameters. A discussion of rationalbehind hyperparameter values will surely improve the paper, and may provide insights into effects of each hyperparameter.


[1] Shekhar, S., Pai, D., & Ravindran, S. (2020, April). Entity resolution in dynamic heterogeneous networks. In Companion Proceedings of the Web Conference 2020 (pp. 662-668).

**Questions:**

How is the TRW-GCN different from the method presented in [1]? Can you include a salesman matrix comparing the given method and other dynamic graph representation learning methods across different characteristics?

---

### Note · Authors · 2024-12-27

**Comment:**

We thank the reviewers for their constructive feedback and will take it into account for any future submissions.

**Withdrawal Confirmation:**

I have read and agree with the venue's withdrawal policy on behalf of myself and my co-authors.